# Diagnosis and Surgical Management for Advanced Pancreatic Cancer Requiring Vascular Resection

**DOI:** 10.3390/diagnostics16010102

**Published:** 2025-12-28

**Authors:** Solonas Symeou, Evangelos D. Lolis, Georgios K. Glantzounis

**Affiliations:** HPB Unit, Department of Surgery, Faculty of Medicine, University of Ioannina and University Hospital of Ioannina, 45110 Ioannina, Greece; md07011@uoi.gr (S.S.); vlolis@uoi.gr (E.D.L.)

**Keywords:** pancreatic adenocarcinoma, PDAC, borderline resectable, locally advanced, vascular reconstruction, venous resection, arterial resection

## Abstract

Pancreatic ductal adenocarcinoma (PDAC) remains one of the most aggressive malignancies, with overall survival outcomes that have improved only modestly in recent years. Careful preoperative evaluation is essential for defining resectability and planning surgery. Modern imaging modalities, including high-resolution, contrast-enhanced CT, MRI and endoscopic ultrasound, provide a detailed assessment of vascular involvement and allow accurate staging according to various international criteria and consensus statements. In borderline and locally advanced cases, neoadjuvant therapy can aid in downsizing the tumor and increasing the likelihood of achieving negative margin resection (R0), offering long-term survival along with quality of life. When vascular invasion limits resectability, venous resection and reconstruction may permit an R0 resection in patients with borderline resectable disease that is both technically operable and physiologically tolerable for the patient. Arterial resection, however, remains controversial and is rarely justified because of its limited perioperative and survival benefits. Arterial divestment has emerged as an interesting alternative, allowing tumor clearance while avoiding full arterial reconstruction. Vascular reconstructions can be achieved through venorrhapy, end-to-end anastomosis, or segmental replacement using either autologous or synthetic grafts. With the advances in neoadjuvant treatment, the appropriate selection of candidates for vascular resection significantly increases the resectability rate, offering long-term survival along with satisfactory quality of life. In this review, a detailed literature review is performed regarding the best strategies in the diagnosis and surgical management of patients with borderline resectable and locally advanced pancreatic cancer requiring vascular resection.

## 1. Introduction

Pancreatic ductal adenocarcinoma (PDAC) is a high-burden malignancy with rising incidence and remains among the leading causes of cancer-related death worldwide [1]. Most patients present with regionally advanced or metastatic disease at diagnosis, such that curative options are limited and only about 10 to 20% are candidates for upfront resection [2,3,4].

Despite these constraints, therapeutic advances over the past decade have yielded remarkable improvements [5,6]. Adjuvant therapy improves outcomes after resection, with durable benefit confirmed at long-term follow-up [7,8,9]. Borderline resectable (BR) and locally advanced (LA) PDAC constitute a substantial proportion of cases, estimated at approximately 40%, for which neoadjuvant therapy (NAT) is now a cornerstone of care [10]. Across multiple studies, NAT increases the likelihood of resectability and margin-negative resection (R0) and is associated with improved survival compared with upfront surgery or palliative chemotherapy alone [11,12,13,14].

Vascular involvement is a pivotal determinant of operability and oncologically acceptable resection in patients with PDAC [15,16]. It is explicitly incorporated into resectability definitions and treatment algorithms from the International Study Group for Pancreatic Surgery (ISGPS) and the National Comprehensive Cancer Network (NCCN), both of which emphasize careful assessment of tumor–vessel relationships when determining operative candidacy [15,17]. Within this framework, venous involvement is commonly managed with resection and reconstruction in specialized units, whereas arterial involvement is approached more cautiously given its substantially higher risk and is undertaken only selectively with strict patient selection [15,18,19,20].

## 2. Materials and Methods

This narrative review synthesizes contemporary evidence on PDAC with vascular involvement, focusing on imaging-based diagnosis and resectability assessment, including the evaluation of venous and arterial invasion, the integration of multimodality treatment strategies, and the technical and clinical outcomes of pancreatectomy with vascular resection and reconstruction. Literature identification followed a focused, non-systematic approach. PubMed and Scopus were queried using predefined search strings, supplemented by reference-list screening and review of current guidelines from major professional societies, including the NCCN and the ISGPS. One author screened PubMed and a second screened Scopus, and potentially relevant records were compiled.

Eligible literature comprised primary and secondary studies aligned with the aims of this review. Abstracts, conference presentations, and non-English publications were excluded. Studies restricted to pancreatic cystic lesions or periampullary tumors were also excluded. Mixed cohorts were included only when patients undergoing major pancreatectomy with concomitant vascular resection for non-PDAC indications accounted for less than 15% of the study population. Final inclusion was determined by consensus among the authors. The Appendix A provide the search strings and a qualitative flow diagram summarizing the study selection.

## 3. Diagnosis

Early diagnosis is crucial because the window for curative-intent surgery narrows once vascular involvement or microscopic dissemination has occurred [21,22]. Two factors drive the delay. Early PDAC is often clinically silent, and even high-resolution imaging can miss small or isoattenuating foci until symptoms or complications arise [22,23,24]. High clinical suspicion and timely surgical triage should follow the documentation of key red flags in a patient’s medical history. These include new or worsening mid-life diabetes, painless jaundice, steatorrhea, itching, weight loss, tobacco exposure, and prior pancreatitis [25]. On physical examination, cholestatic stigmata, a painless palpable gallbladder, or vague epigastric pain radiating to the back should also raise suspicion [25].

Baseline tests include complete blood count, liver function tests (TBL, DBL, ALP, γ-GT, ALT, AST), and fasting glucose or HbA1c [26,27]. Abnormalities in these parameters should prompt further evaluation with appropriate imaging to assess for biliary obstruction and a potential underlying PDAC. Carbohydrate antigen 19-9 (CA 19-9) can aid tumor characterization and treatment monitoring, but it is not a screening test. In the setting of biliary obstruction, values should be interpreted after biliary decompression, and clinicians should recognize that Lewis^a^ antigen-negative patients do not secrete CA 19-9 [28,29,30]. Across meta-analyses in symptomatic cohorts, pooled sensitivity and specificity of CA 19-9 in diagnosing PDAC range from 72% to 80% and from 80% to 86%, respectively, underscoring its adjunctive rather than standalone role [31,32,33]. Two additional meta-analyses of 10 and 41 studies reported that higher preoperative CA 19-9 levels were associated with occult metastases (mean difference = 904.4, 95% CI: 642.08–1166.74) and that higher postoperative levels were associated with shorter survival (HR = 1.79, 95% CI: 1.58–2.01), respectively [34,35].

Transabdominal ultrasound is inexpensive and widely available, and is therefore frequently used as an initial triage modality in patients presenting with jaundice [36]. However, it should not delay cross-sectional imaging when clinical suspicion is high, even when initial findings are suggestive of PDAC. Diagnostic performance is strongly operator-dependent and can be limited by patient habitus and bowel gas [36]. Meta-analyses and contemporary series report sensitivity in the range of 67.7% to 88% and specificity of approximately 75% to 94%, which remains inferior to cross-sectional modalities for definitive workup [22,37,38,39].

Pancreas-protocol computed tomography (CT), incorporating pancreatic parenchymal and portal venous phases with multiplanar reconstructions, remains the primary modality for diagnosis and assessment of resectability, providing systematic vascular mapping to guide operative planning in most patients [22,40,41]. Typical imaging features include a hypoenhancing mass, abrupt main pancreatic duct cutoff with distal gland atrophy, the double duct sign, and a precise description of tumor relationships to the portal vein (PV), superior mesenteric vein (SMV), superior mesenteric artery (SMA), common hepatic artery (CHA), and celiac axis (CA) [22,40,42,43]. Contemporary data demonstrate that CT detects PDAC with a sensitivity near 90% and a specificity of around 89% [22]. For resectability prediction, the summary positive predictive value of CT is approximately 81%, such that about one in five patients initially thought to be resectable on imaging are found to have occult metastatic or otherwise unresectable disease at exploration [44].

Endoscopic ultrasound (EUS) provides the highest spatial resolution for the pancreas and peripancreatic region and is preferred for tissue acquisition when histologic confirmation is required, particularly before the initiation of NAT [45,46]. It typically complements CT by improving the detection of small or isoattenuation tumors and refining locoregional staging, particularly for lesions smaller than 3 cm [46,47,48,49]. Ancillary techniques can further enhance its performance. Contrast-enhanced EUS may improve diagnostic accuracy by better delineating viable tumor and facilitating more precise needle targeting, although the magnitude and consistency of this effect are not yet definitive [50,51]. Elastography may strengthen the assessment of vascular involvement by enabling a more objective appraisal of the tumor–vessel interface. In this context, the preservation of a distinct intervening, band-like layer between the tumor and the vessel argues against vascular wall invasion, whereas the disruption or absence of this layer supports direct infiltration when vessel contact is otherwise indeterminate [52]. In unresectable disease, EUS can also enable timely palliation through adjunct interventions such as celiac plexus neurolysis and, in selected cases, intratumoral ethanol ablation [53]. These techniques are increasingly integrated into multimodal palliative care pathways, reflecting their potential to improve analgesia and reduce opioid requirements. Nevertheless, reported clinical benefit is variable, and robust criteria for patient selection remain incompletely defined.

Magnetic resonance imaging (MRI) serves as an alternative to CT when iodinated contrast is contraindicated or lesion conspicuity is limited [40]. For assessing local vascular involvement, available comparative evidence demonstrates that MRI is not superior to high-quality CT for operative decision-making [54,55]. Accordingly, MRI should be reserved for cases in which CT findings are equivocal, rather than used routinely. Its most consistent added value in contemporary practice is metastatic staging, particularly in the liver, where it is more sensitive than CT for detecting occult metastases and can directly influence resectability assessment and treatment planning [56]. Beyond standard MRI sequences, magnetic resonance cholangiopancreatography (MRCP) provides incremental diagnostic utility by noninvasively characterizing the pancreaticobiliary ductal anatomy, identifying subtle ductal abruptions, and offering a reliable non-invasive alternative to endoscopic retrograde cholangiopancreatography (ERCP) for preprocedural ductal mapping [57].

Fluorodeoxyglucose positron emission tomography (FDG-PET/CT) is not a first-line test for detecting PDAC or defining local resectability [57]. It is primarily used for whole-body staging before and after NAT in biologically high-risk disease or when conventional imaging identifies indeterminate extrapancreatic findings suspicious for metastases [50,58]. Changes in FDG uptake on serial PET/CT may serve as a useful indicator of treatment response and can inform decisions regarding whether to proceed with surgical exploration and resection [58].

In PDAC with vascular involvement, staging laparoscopy refines anatomic assessment, clarifies operability, and helps avoid nonbeneficial laparotomy [59,60]. Current NCCN guidelines (2025) advise considering staging laparoscopy for inconclusive preoperative imaging or high-risk features such as markedly elevated CA 19-9 (>100 U/mL or >250 U/mL), low-volume ascites, tumors in the body or tail of a size > 3 cm, borderline resectable anatomy, or suspicious bile duct nodes [17]. After NAT, mild perivascular soft-tissue stranding is common and should not, by itself, preclude exploration, particularly when CA 19-9 is stable or declining, cross-sectional imaging shows no radiographic progression, and PET-CT demonstrates a meaningful reduction in post-treatment metabolic activity.

Table 1 summarizes the diagnostic modalities and their clinical indications for PDAC assessment, together with key imaging findings. Figure 1 complements this overview by outlining a practical diagnostic algorithm for PDAC.

### Definitions of Vascular Involvement and Resectability

As aforementioned, operability is determined largely by the extent and pattern of the tumor–vessel interface. Venous relationships are addressed first, using a standardized nomenclature that stratifies patients by the extent of circumferential tumor vessel contact [17]. Abutment is contact of 180 degrees or less (≤180°) and encasement is contact of greater than 180 degrees (>180°). This terminology should be applied consistently by clinicians and within multidisciplinary tumor boards (MDTs) [17].

Venous invasion on pancreas-protocol CT is suggested by extensive tumor–vessel contact accompanied by secondary venous morphologic changes. Supportive features include venous contour irregularity, focal luminal narrowing, segmental occlusion, the teardrop sign, and an intraluminal filling defect, particularly when it is contiguous with the primary tumor or demonstrates enhancement, favoring tumor thrombus over bland thrombosis [21,22]. Arterial invasion should not be diagnosed on abutment alone. In contrast, arterial encasement is a strong predictor of true invasion, with specificity approaching 100% and sensitivity near 80% [22]. Additional features that increase confidence include focal narrowing, contour irregularity, a tapered near-occlusive “string sign,” and a perivascular “halo” sign [22].

For the SMV and PV, resectable disease includes no tumor contact or abutment without secondary venous deformity. BR disease includes abutment or encasement accompanied by focal narrowing, contour irregularity, or short-segment thrombosis when suitable proximal and distal targets exist for safe and durable reconstruction. Tumor contact with the inferior vena cava is also classified as BR. LA venous disease is defined as complete occlusion of the SMV or PV that is not amenable to resection with primary reconstruction at presentation [17].

Arterial criteria are deliberately stringent because achieving an R0 resection is less probable and technically more challenging when arterial planes are compromised, and safe clearance of the SMA and CA may not be feasible without prohibitive risk [17]. For tumors of the pancreatic head or uncinate process, resectable disease requires no measurable tumor contact with the CA, SMA, or CHA. BR disease includes short-segment abutment of the CHA without extension to the CA or the hepatic artery (HA) bifurcation, and abutment of the SMA or limited abutment of a variant HA when safe reconstruction is feasible. LA disease is present with encasement of the SMA or CA, as clearance of the arterial plane is unlikely to yield negative margins [17].

For tumors of the pancreatic body or tail, resectable disease likewise requires no measurable tumor contact with the CA, SMA, or CHA. BR disease includes abutment of the CA when divestment or reconstruction is feasible in high-volume centers. LA disease includes encasement of the SMA or CA, as well as the combined involvement of the CA and the aorta.

Table 2 and Table 3 provide a synopsis of the current NCCN guidance for determining PDAC operability in the setting of venous or arterial involvement.

## 4. Management of PDAC with Vascular Involvement

Resectable PDAC is generally managed with upfront pancreatectomy followed by contemporary adjuvant systemic therapy, conventionally delivered over approximately 6 months. In medically fit patients with an Eastern Cooperative Oncology Group (ECOG) performance status of 0 to 1, a FOLFIRINOX-based regimen is preferred and is typically administered as 12 biweekly cycles [17]. Available clinical evidence in resected PDAC supports benefit in both node-negative (pN0) and node-positive (pN1) subgroups [17,61,62]. Across two randomized trials, FOLFIRINOX improved median overall survival (OS) to approximately 54 months versus 35 months with gemcitabine, with benefit maintained in pathological advanced disease (pT3 to pT4) and pN1 status [61,62]. Gemcitabine plus capecitabine also improves survival compared with gemcitabine alone, although the absolute gain is modest at 28 versus 25.5 months [63]. By contrast, adjuvant nab-paclitaxel plus gemcitabine did not significantly improve disease-free (DFS) or OS relative to gemcitabine monotherapy [64]. The CONKO-001 trial further confirmed the inferiority of surgery alone, showing improved DFS and long-term OS with adjuvant gemcitabine versus observation after R0 or R1 resection [65,66]. Pooled comparative analyses are concordant, consistently ranking FOLFIRINOX-based strategies highest for DFS and OS and showing no advantage of gemcitabine plus nab-paclitaxel over gemcitabine plus capecitabine [67,68]. Gemcitabine-based therapy is generally reserved for frailer patients or for those in whom comorbidities or limited functional status, including ECOG 2, preclude intensive treatment. When selected, it is typically delivered as six cycles administered every 4 weeks [17].

NAT is the standard of care for BR PDAC. It is also increasingly used for anatomically resectable tumors with biologically high-risk features, including elevated CA 19-9, suspicious regional lymph nodes, or large primary tumors, in accordance with current NCCN guidelines despite heterogeneity in the supporting evidence [17,69,70,71]. Contemporary strategies are commonly structured around an approximately 6-month course of systemic therapy. In routine practice, this is delivered as 12 cycles of FOLFIRINOX or 6 cycles of a gemcitabine-based regimen, with sequencing tailored to patient fitness and tolerability, as outlined above [17,72]. In many institutions, the preoperative component spans 2 to 4 months and typically includes 4 to 6 cycles of FOLFIRINOX, followed by postoperative chemotherapy to complete 12 total cycles, usually an additional 6 to 8 cycles. A similar perioperative approach is used for gemcitabine-based regimens, most often two to four cycles preoperatively and two to four cycles postoperatively [17,73,74]. Interestingly, total NAT is increasingly adopted. Under this approach, the entire intended course of systemic chemotherapy is completed preoperatively, such that postoperative adjuvant chemotherapy is not planned, particularly when completion of therapy after major pancreatic resection is considered unlikely [17]. Clinical trial data support systemic therapy as the backbone of preoperative management. In the ESPAC 5F trial, short-course NAT strategies improved 1-year OS compared with immediate surgery. One-year OS was 39% with immediate surgery and ranged from 60% to 84% across NAT arms, with the highest estimate in the FOLFIRINOX arm [75]. In Alliance A021501, neoadjuvant-modified FOLFIRINOX (mFOLFIRINOX) alone achieved superior 18-month and median OS and higher R0 resection rates compared with mFOLFIRINOX followed by hypofractionated radiotherapy, preoperatively. The chemoradiation arm was closed early, suggesting no clear benefit from routine radiotherapy intensification when an effective systemic chemotherapy regimen is administered [76]. A meta-analysis of 1570 patients similarly reported longer pooled median OS with neoadjuvant chemotherapy alone than with chemoradiotherapy—26 versus 18 months—despite higher pooled R0 resection rates with chemoradiotherapy of approximately 83% versus 70% [77]. However, this survival difference was not observed when radiation was preceded by at least five induction cycles, supporting a strategy in which radiotherapy is reserved for selected patients after adequate systemic exposure [77].

As in borderline resectable PDAC, locally advanced PDAC is typically treated with several months of combination induction chemotherapy, with regimen selection and dose intensity tailored to performance status and comorbidity burden, usually two to six cycles. This strategy enables conversion to resection in a minority of patients, and long-term benefit is primarily determined by tumor biology and depth of response [78,79,80]. In patients without distant metastases, radiotherapy may be considered selectively to enhance durable local control, either as consolidation after induction chemotherapy or in carefully selected patients who are not candidates for multi-agent induction therapy, delivered as conventionally fractionated chemoradiation over 5 to 6 weeks (45–56 Gy) or stereotactic body radiotherapy over 1 to 2 weeks (30–45 Gy) [17].

Response assessment should be performed using a standardized, multimodal approach in BR PDAC and, importantly in LA PDAC, following induction therapy [17]. Candidacy for surgical, preferably laparoscopic, exploration is supported by preserved functional status with adequate tolerance of therapy, favorable CA 19-9 kinetics assessed after bilirubin normalization, and restaging pancreas-protocol CT (or MRI) showing no interval progression and no new metastatic disease [17]. In both settings, substantial radiographic downstaging is uncommon [17]. Clinically meaningful response is therefore more reliably reflected by radiologic non-progression or only modest tumor shrinkage accompanied by a marked CA 19-9 decline and reduced metabolic activity on PET/CT [17]. When these criteria are met and an R0 resection remains technically achievable based on surgical assessment, pancreatic resection with vascular resection and reconstruction should be strongly considered following MDT discussion, including in selected patients with LA disease [17]. This approach is supported by pooled analyses showing that patients who respond to induction therapy and proceed to resection achieve better survival than those managed with continued palliative systemic therapy [79,81]. In aggregate, these data argue for systematic restaging after induction therapy, timely referral to high-volume hepatopancreatobiliary (HPB) centers, and consideration of exploration in carefully selected patients with non-progression and favorable biomarker and metabolic response profiles [79,81,82].

In patients with pancreatic head tumors and clinically significant obstructive jaundice, endoscopic biliary decompression with ERCP-guided transpapillary stent placement is the preferred approach, relieving hyperbilirubinemia and cholestasis while avoiding surgical bypass and enabling timely pancreaticoduodenectomy (PD) when indicated [83,84]. In resectable PDAC scheduled for upfront resection, routine preoperative biliary drainage (PBD) is not recommended, given higher rates of infectious complications and overall morbidity without a survival advantage compared with proceeding directly to surgery [85,86,87]. As such, PBD should be reserved for acute cholangitis, severe jaundice with organ dysfunction, substantial comorbidity requiring optimization, or an anticipated delay to operation [86,88]. When NAT is planned, PBD should be performed in nearly all jaundiced patients, including those with BR and LA PDAC as well as selected resectable tumors with aggressive biological features. In this setting, biliary stenting facilitates bilirubin normalization and reduces biliary complications during systemic therapy, serving as a practical bridge to definitive resection when feasible or, when appropriate, palliative management [17].

Table 4 summarizes the therapeutic rationale for pancreatic ductal adenocarcinoma and outlines the initial management strategy, follow-up approach, and key considerations for surgical management.

## 5. Surgical Management

Achieving an R0 resection remains the central objective in PDAC with suspected or confirmed vascular involvement and is generally approached with en-bloc pancreatectomy incorporating a planned vascular resection and reconstruction strategy [89,90]. The extent of pancreatic parenchymal resection is guided primarily by tumor location. PD is performed for lesions arising in the head or uncinate process, whereas distal pancreatectomy (DP) is indicated for lesions of the body or tail. Total pancreatectomy (TP) may be selectively considered when an R0 resection is unlikely to be achieved with partial pancreatectomy [90,91]. It may also be appropriate in the setting of extensive vascular reconstruction or in carefully selected older patients to reduce the risk of pseudoaneurysm formation associated with postoperative pancreatic fistula or anastomotic leak [92]. In high-volume centers, venous resection (VR) has become an established adjunct to pancreatectomy for BR and for technically operable LA PDAC, and is endorsed by multiple guidelines [17,90]. By contrast, the role of arterial resection (AR) remains uncertain. Available series report higher morbidity without a consistent survival advantage, and AR should therefore be restricted to carefully selected patients in centers with appropriate expertise [17,90,93,94].

Nodal clearance should follow the ISGPS definition of standard lymphadenectomy and be adapted to tumor location. Extended dissections are discouraged because they do not improve survival and are associated with increased morbidity [95]. In this context, patients requiring vascular resection should undergo oncologic pancreatectomy with en- bloc vascular resection and reconstruction, together with standardized lymphadenectomy according to an established nodal template [17,90,95].

### 5.1. Venous Resections and Reconstructions in PDAC

In a nationwide cohort from the Netherlands including 1311 PDs across 18 hospitals, reported VR in 27% of cases, including 17% wedge and 10% segmental resections [96]. Consistent with these national data, a prospective international registry including 3926 pancreatic operations across 167 centers documented 565 vascular resections. Of these, 444 were VRs, accounting for 79% of all vascular resections [97]. Collectively, these data indicate that VR and reconstruction are an integral component of contemporary pancreatic surgery and account for the majority of vascular procedures, underscoring the need for standardized operative definitions and consistent outcome reporting to support the ongoing refinement and periodic updating of clinical practice guidelines, and to define training, credentialing, and expectations for these technically demanding operations.

The ISGPS proposes a classification that provides a common language for venous resection and correlates with technical complexity and clinical risk. Type 1 refers to tangential excision with primary venorrhaphy, type 2 to tangential excision with patch venoplasty, type 3 to segmental resection with primary end-to-end anastomosis, and type 4 to segmental resection with an interposed conduit requiring two anastomoses (proximal and distal) [98]. Appendix A provides a summary of the ISGPS VR classification and reconstruction.

The operative sequence prioritizes early exposure and vascular control of the portomesenteric axis to secure venous inflow and outflow and tailor reconstruction to the anticipated defect. This approach aims to reduce the risks of stenosis and thrombosis. When collateralization is extensive or portal inflow is tenuous, temporary portomesenteric or portocaval shunting, catheter bypass, or both may be used to re-establish dependable hepatic inflow before tumor mobilization and to reduce ischemia time during reconstruction [99,100].

#### 5.1.1. Oncologic Outcomes of Pancreatectomy with Venous Resection

Across comparative analyses, VR appears to function primarily as a surrogate for more advanced disease rather than an independent determinant of survival. In a meta-analysis of 27 studies including 9005 patients, Giovinazzo et al. reported lower R0 resection rates and inferior 1-, 3-, and 5-year OS after PV or SMV resection compared with standard resection (HRs = 1.23, 1.48, and 3.18, respectively), accompanied by a shorter median OS of 14.4 versus 19.5 months. On this basis, the authors concluded that the observed survival decrement predominantly reflects adverse tumor biology within VR cohorts rather than the reconstructive procedure itself [101]. Zwart et al. reached a concordant interpretation, also demonstrating lower 1-, 3-, and 5-year survival in VR versus non-VR cohorts (55.8% vs. 60.6%, 18.3% vs. 24.3%, and 13.0% vs. 20.0) while endorsing VR when required to achieve margin clearance [102]. A recent systematic review further contextualized these outcomes, reporting median OS of 12 to 28 months after VR across contemporary series, which provides a pragmatic survival range for these higher-risk populations [103].

From a surgical oncology perspective, the attainment of an R0 resection remains the most consistently reproducible determinant of long-term survival. In this context, Yu et al. confirmed lower R0 rates and reduced 5-year survival with VR compared with standard procedures, while demonstrating that margin status strongly stratified outcomes within the VR group. Specifically, 2- and 5-year OS were significantly higher when R0 was obtained (ORs = 2.93 and 4.20, respectively) [104]. In pooled analyses, Wang et al. synthesized 7567 patients and similarly reported a reduced R0 frequency in VR cohorts (60.5% versus 68.7%) with lower 5-year survival (12.7% versus 15.4%, *p* = 0.0001), while emphasizing that oncologic adequacy remains central and that VR can be performed safely in experienced centers [105]. Institutional single-center experiences provide additional context for these associations. Tsiotos et al. reported a median OS of 24 months after pancreatectomy with portomesenteric vein resection for BR and LA PDAC, with OS extending to 32 months in patients with ECOG 0, underscoring the importance of patient fitness and selection [106]. Belloti et al. likewise observed lower R0 rates after VR versus standard PD (58.0% versus 82.1%, *p* = 0.0001) together with shorter median OS (21 versus 30 months, *p* = 0.023), supporting the interpretation that inferior survival in VR cohorts is largely mediated by lower R0 rates in the setting of more advanced local disease and adverse tumor biology [107].

Although conclusions are tempered by confounding related to case selection and variation in operative technique and follow-up, VR is appropriately regarded as an oncologic adjunct when venous involvement would otherwise preclude complete clearance. Benefit is most credible when R0 is realistically achievable, reinforcing careful selection based on physiologic reserve and response to NAT. Accordingly, VR should be concentrated in high-volume centers with established expertise in portomesenteric reconstruction and where care is consistently delivered within an MDT setting.

#### 5.1.2. Reconstruction Technique, Patency and Thrombosis

Durable portomesenteric inflow after PD requires a tension-free reconstruction that preserves luminal diameter and flow, with the technique tailored to defect geometry and length. For limited defects in which native caliber can be maintained, tangential venorrhaphy with transverse suturing helps preserve vessel diameter and reduces shear-related stenosis, supporting an anatomy-preserving repair strategy [108]. In a 90-patient, technique-stratified cohort, late patency was 100% after tangential venorrhaphy and 100% after primary end-to-end anastomosis. By contrast, thrombotic complications clustered after prosthetic conduit interposition, which accounted for 44% of thrombotic events (*p* = 0.001) [108]. A separate study reported a similar patency gradient at 1-year follow-up, favoring simpler repairs over prosthetic conduits, with patency rates of 100% after venorrhaphy, 81.8% after end-to-end anastomosis, and 66.7% after prosthetic interposition grafting [109].

For broad, partial-circumference defects in which primary closure would predictably narrow the lumen, patch venoplasty preserves venous caliber and can be performed using immediately available autologous tissues with favorable infection profiles, including parietal peritoneum and the falciform ligament [110,111,112]. In a dedicated HPB series, parietal peritoneum achieved 100% patency when used as a lateral onlay patch, underscoring its utility as a readily available graft option in urgent settings with minimal donor-site morbidity [111]. Falciform ligament patch venoplasty has also demonstrated excellent patency in retrospective cohorts, with limited need for antiplatelet or anticoagulant therapy [113,114]. These tissues can also be fashioned into tubularized constructs, although overall patency data for such configurations remain limited and inconsistently reported.

Primary end-to-end anastomosis provides excellent patency for short segmental, circumferential defects when venous mobilization permits a tension-free, well-aligned repair, and it is generally appropriate for gaps shorter than 2 to 3 cm [108]. In contrast, a planned venous resection length of at least 31 mm, significant caliber mismatch, or any residual tension should favor interposition grafting, although longer primary repairs have been reported in highly selected cases at experienced centers [101,115,116]. When a conduit is required, graft material strongly influences durability. Autologous and allogeneic venous conduits are associated with lower thrombosis rates and superior patency relative to synthetic grafts, with reported early and overall thrombosis of 5.6% and 11.7% for autologous vein and 2.5% and 6.2% for venous allograft, compared with 7.5% and 22.2% for synthetic grafts, supporting a vein-first reconstructive strategy [117]. This preference is biologically plausible given greater biocompatibility and more physiologic compliance, with infection risk that appears not inferior to synthetic grafts within the portomesenteric circulation [113,118]. Nevertheless, venous conduits may be limited by patient fitness, donor-site morbidity considerations (e.g., lower-limb edema, renal insufficiency), and, for allografts, restricted availability, such that prosthetic grafts remain an appropriate alternative when other options are not feasible. In this setting, acceptable outcomes depend on precise diameter matching and a rigorously tension-free configuration to mitigate thrombosis risk [116,117,118,119].

### 5.2. Arterial Resections and Reconstructions in PDAC

When arterial involvement is suspected, early anatomy-based decision-making is critical. An artery-first strategy prioritizes early exposure of the relevant arterial segments to assess abutment or encasement and obtain proximal control, enabling timely determination of whether an R0 resection is achievable with arterial preservation or whether arterial reconstruction is necessary [120,121,122]. The specific dissection route should be individualized to tumor topography rather than dictated by a fixed operative sequence.

When a suitable periarterial plane is present, arterial divestment can achieve oncologic clearance while avoiding graft reconstruction and the associated technical complexity and postoperative surveillance requirements [123,124]. Accordingly, AR should be confined to high-volume programs with dedicated vascular expertise and established rescue pathways, given the high-consequence risks of hemorrhage, ischemic complications, and graft failure and the variable, biology-dependent benefit.

#### 5.2.1. Oncologic Outcomes of Pancreatectomy with Arterial Resection and Arterial Divestment

At present, routine AR in PDAC is not clearly justified. Reported perioperative and oncologic outcomes vary substantially across centers, and any apparent improvement in R0 resection rates is inconsistent. Even when higher R0 rates are achieved, this has not reliably translated into improved OS. Accordingly, the current evidence base remains insufficient to support guideline endorsement of routine AR, leaving patient selection and intraoperative decision-making particularly challenging.

In pooled data from 5465 patients, AR increased the likelihood of an R0 resection (RR = 3.11, 95% CI 1.65–5.86) without a significant increase in overall morbidity, yet mortality was higher than with standard resections (6.4% vs. 1.8%, *p* = 0.036). These findings indicate that any margin advantage may be offset by perioperative risk when AR is applied broadly and outside tightly standardized pathways [125]. More favorable outcomes have been reported when AR was evaluated in patients treated with NAT. In a meta-analysis limited to major AR performed after NAT, pooled estimates included an R0 rate of 79%, morbidity of 51%, and mortality of 2%, with survival rates of 92.3% at 1 year, 64.8% at 2 years, 51.6% at 3 years, and 14% at 5 years [126]. Although these aggregate data do not establish causality, they are consistent with the hypothesis that NAT may improve the risk–benefit balance of major vascular surgery through systemic disease control and, potentially, by preferentially advancing treatment-responsive tumors to resection.

In line with this, a 203-patient comparative series spanning 2010–2021 reported improved outcomes in the later period (2016–2021), coinciding with widespread NAT adoption, with median OS increasing from 26.0 to 48.2 months and major morbidity decreasing significantly [124]. Within subgroup analyses, median OS after AR rose from 22.0 to 45.1 months, but this represented a numerical, non-significant trend. In contrast, a statistically significant improvement was observed with arterial divestment, with median OS increasing from 26.0 to 57.1 months, supporting this approach when oncologic clearance can be achieved along a true peri-adventitial plane [124]. Concordantly, in a cohort of 259 vascular resections for BR and LA PDAC, AR was undertaken in a small, highly selected subset of 19 patients with presumed arterial involvement [127]. Pathological R0 rates were similar with versus without AR (37.5% vs. 21.7%, *p* = 0.17), whereas mortality trended higher in the AR, including isolated AR and AR with concomitant VR (21.1% vs. 5.0%) [126]. Notably, transmural invasion was confirmed in only 6 of 19 AR cases, suggesting that suspected arterial infiltration is frequently overcalled and may expose some patients to potentially avoidable AR and its associated morbidity [127]. For distal pancreatic tumors, the absence of a clear OS benefit with DP-CA reconstruction compared with standard DP is likely influenced by tumor biology, as these lesions are often diagnosed later and may harbor occult micrometastatic disease, limiting incremental survival gains despite technically successful R0 resection [128].

Ultimately, AR is typically reserved for patients with more extensive vascular involvement and is frequently combined with VR, which increases operative complexity, reconstructive demands, and postoperative morbidity. These considerations support a disciplined, pathway-driven strategy in which arterial divestment is used when oncologic clearance is achievable, and AR is limited to short, technically straightforward reconstructions in rigorously selected patients treated in specialized multidisciplinary programs with the requisite vascular expertise.

#### 5.2.2. Patency and Thrombotic Events Following Arterial Reconstruction

In a vascular surgery–led cohort of 111 arterial interposition reconstructions using a mix of autologous arterial and venous conduits, allografts, and synthetic grafts, reconstruction-related vascular complications were not infrequent. Events attributable to the reconstruction occurred in 11% of cases and included graft thrombosis and clinically significant stenosis requiring reintervention. Overall, nine patients underwent vascular reinterventions, highlighting a meaningful postoperative failure burden even in experienced programs [129].

A focused series of 108 HA resections further illustrates the thrombotic burden and underscores the practical importance of both reconstruction strategy and conduit selection. Complete HA occlusion occurred in 18%, with events clustering in the first postoperative months, supporting an early period of heightened surveillance [130]. Occlusion rates varied by technique, occurring in 22.7% after interposition grafting compared with 9.5% after end-to-end anastomosis, which supports primary reconstruction when technically feasible [130]. When interposition grafting is required, conduit choice appears to be clinically relevant, with complete occlusion rates of 15% for arterial interposition grafts, 44% for venous grafts, and 60% for synthetic grafts, reinforcing a hierarchy that favors autologous arterial conduits over venous or prosthetic substitutes when possible [130].

While these observations parallel patterns seen in VR, they are particularly pertinent at the arterial interface because the arterial wall, supported by a thick tunica media, is relatively resistant to transmural invasion. Consequently, arterial invasion inferred from imaging alone may be overstated in a meaningful proportion of patients. Operative decision making should therefore be anchored in exploratory laparoscopy and careful intraoperative assessment to confirm true arterial involvement and to determine whether a safe peri-adventitial plane can be developed. In the setting of expanded NAT and contemporary use of radiotherapy, arterial divestment should be the default escalation strategy for borderline arterial contact when arterial integrity can be preserved. Arterial reconstruction remains an option, but it should be reserved for unequivocal, non-salvageable segments when an R0 resection cannot otherwise be achieved.

## 6. Antithrombotic Practice

Postoperative antithrombotic management after pancreatectomy with VR and AR remains highly variable, and no evidence-based standard protocol has been established [131,132,133]. Pancreatectomy is a major abdominal operation with increased thrombotic risk, which is further amplified when PDAC is the operative indication. Available data derive largely from retrospective single-center cohorts and a limited number of systematic reviews, with marked heterogeneity in surgical technique, reconstruction type, antithrombotic exposure, and imaging surveillance, which constrains inference and limits generalizability [131,134,135]. Accordingly, management is individualized, balancing thrombosis prevention against hemorrhagic complications, including post-pancreatectomy hemorrhage and delayed bleeding from arterial pseudoaneurysms [135].

In current clinical care, postoperative antithrombotic strategies range from unfractionated heparin or low-molecular-weight heparin administered at prophylactic or therapeutic doses to warfarin, aspirin, and clopidogrel. Formal institutional protocols are often lacking, and prescribing is frequently left to surgeon discretion. Decisions are guided by patient-specific thromboembolic indications, including atrial fibrillation, prosthetic heart valves, and prior thromboembolism. Operative and technical factors also influence management, including operative duration, resection length, reconstruction method, and use of a synthetic graft. In a recent systematic review, anticoagulation was used more often after tangential reconstruction and end-to-end anastomosis and in the setting of synthetic graft interposition, yet outcomes were not consistently improved compared with no anticoagulation [134]. Early thrombosis within 30 days, which is directly associated with increased mortality, occurred more frequently with anticoagulation at roughly 5% to 7% versus 2% to 3%, although statistical significance varied across studies. Mortality and overall morbidity were similar at 5% versus 6% and 47% versus 40%, respectively. Bleeding rates also varied widely, reported at 7% to 8% with anticoagulation versus 7% to 22% without anticoagulation, with study-dependent significance [134,135].

Direct oral anticoagulants are seldom described after VR or AR and, when mentioned, are usually continued for established indications rather than initiated to preserve graft patency, leaving their role in this setting incompletely defined [135,136,137]. Overall, current evidence does not justify routine therapeutic-intensity anticoagulation after pancreatic vascular reconstruction. A practical strategy is to provide postoperative prophylaxis individualized to reconstruction complexity, early postoperative imaging confirmation of flow, and patient-specific thrombotic and bleeding risk. Escalation to therapeutic anticoagulation or intensified antiplatelet therapy should be reserved for clearly increased risk, such as complex reconstructions, flow limitation on imaging, or established thrombosis, with explicit consideration of post-pancreatectomy hemorrhage through multidisciplinary decision making [131,134,135].

## 7. Discussion

Pancreatic ductal adenocarcinoma is a lethal malignancy, with both incidence and mortality projected to increase over the coming decades [1]. In this setting, where care pathways are complex and time-sensitive, outcomes reflect not only tumor biology, but also the quality, coordination, and timeliness of care delivery. Centralization to high-volume centers is therefore pivotal, as it concentrates multidisciplinary expertise for coherent treatment planning, improves the consistency of imaging acquisition and interpretation, and provides the technical and perioperative infrastructure required for complex resections that may include vascular reconstruction. Population-level analyses support the clinical value of this organizational model, with high-volume center management associated with lower in-hospital, 30-day, and 90-day mortality, reduced overall morbidity, shorter length of stay, and higher rates of R0 resection [138,139].

Timely diagnosis is paramount because the window for disease eradication is narrow and becomes even more constrained in the presence of vascular involvement or unrecognized dissemination, which are associated with inferior oncologic outcomes. Diagnostic pathways should therefore be designed to minimize the interval between first presentation and high-quality cross-sectional imaging. This emphasis is justified because early PDAC often presents with nonspecific symptoms and newly abnormal laboratory indices rather than distinctive cancer-specific features [23,24,25]. In this context, a structured history and focused examination serve a triage function by identifying symptom patterns, risk factors, and physical signs suggestive of significant pancreaticobiliary pathology that should prompt expedited imaging and early specialist referral [25,26,27].

Accordingly, pancreas-protocol CT should be obtained promptly, as it is the cornerstone study for diagnosis and the primary determinant of initial management [22,40]. Its central value lies in providing a reproducible, surgery-oriented assessment of the tumor–vessel interface, which underpins resectability classification and helps anticipate the technical requirements of resection, including the need for vascular control and, when indicated, reconstruction [22,40,43]. Routine additional testing is not indicated and is best limited to management-relevant questions that remain unresolved after CT, since indiscriminate test “layering” prolongs time to treatment and seldom changes operative planning once vascular relationships and surgical planes are clearly established. Supplementary investigations, when indicated, should be selected through MDT review to maximize incremental diagnostic yield and minimize delay, duplication, and interpretive noise. EUS is indicated to address residual ambiguity in local staging when CT findings are indeterminate, to secure histologic confirmation when necessary, and to facilitate palliative interventions in patients who are not surgical candidates [45,52,53]. MRI is best reserved for cases in which contrast-enhanced CT is contraindicated, most commonly due to a documented prior contrast allergy or advanced renal dysfunction that precludes iodinated contrast administration, or when specific unresolved concerns are likely to alter management, particularly suspected small-volume hepatic or extra-pancreatic metastases that may be more sensitively detected with liver-directed sequences [56].

Historically, PDAC with vascular involvement, encompassing BR and LA disease, was frequently deemed unresectable. This view reflected substantive anatomic constraints and a limited therapeutic window in the pre-NAT era [5,6,98]. Tumor vessel abutment or encasement increased the technical demands of resection, heightened perioperative risk, and reduced the probability of achieving complete oncologic clearance. These anatomic constraints were further compounded by imperfect systemic disease control and the limited ability to reliably discriminate indolent from aggressive tumor biology, which often resulted in early distant progression and diminished the incremental benefit of aggressive local therapy.

Against this backdrop, the integration of NAT, with radiotherapy used selectively, has shifted the paradigm by enabling resection in a subset of patients with durable disease control and by delivering meaningful survival benefit when performed at specialized units with acceptable perioperative morbidity [17,73,74,77]. NAT is best conceptualized as early systemic treatment and an in vivo assessment of biologic behavior, rather than a strategy that reliably produces anatomic downstaging. Consistent with this, post-NAT CT often shows only modest changes in tumor–vessel contact, while therapy-related remodeling at the tumor–vessel interface can further confound assessment of residual invasion [17,21,22,23]. In this setting, FDG-PET/CT may provide complementary information through changes in metabolic activity, and staging laparoscopy can exclude occult metastatic disease, confirm operability, and refine surgical planning [57,58,59,60]. Together, these steps reduce the risk of non-therapeutic laparotomy and support pursuit of curative-intent resection when appropriate.

Contemporary operative decision-making therefore extends beyond anatomic feasibility to the likelihood that surgery will translate into meaningful oncologic benefit. Margin status remains a core determinant of outcomes within a given local stage, underscoring the need to align operative strategy with a realistic expectation of complete tumor removal [89]. Venous involvement is increasingly approached as surgically addressable rather than an automatic contraindication, with operative decisions guided by physiologic reserve and response or stability on NAT and framed within contemporary guideline-based selection [15,17,97,98]. When surgery is pursued, VR is undertaken when necessary to facilitate complete tumor removal, and venous re-vascularization is preferentially executed with the simplest durable configuration, favoring primary repair when tension-free alignment is achievable and reserving interposition grafting for situations in which traction-free approximation cannot be obtained [98,108,109,110,111].

In contrast, arterial resections warrant greater restraint and more stringent patient selection, because their oncologic benefit is less clearly established and the procedural burden is typically higher than that associated with venous reconstruction [140,141,142]. Outcomes reported after AR vary widely across centers, driven by heterogeneity in selection thresholds, operative technique, perioperative care pathways, and study population size, thereby limiting comparability across published series [125,126,127,142]. The indication for AR reflects a more advanced locoregional disease state than isolated venous involvement and may therefore confer a higher risk of both local and systemic relapse, with a corresponding increase in late postoperative mortality [102,103,124,143]. In addition, many candidates reach consideration for AR after prolonged NAT, which may be accompanied by treatment-related deconditioning and can reduce physiologic tolerance for extensive surgery, contributing to the substantial heterogeneity in the observed postoperative outcomes [17,124,126]. AR also commonly coincides with VR, expanding the vascular component of the procedure and increasing technical demands. The requirement for multiple reconstructions, often during a prolonged operation, can compound perioperative risk and is associated with a higher likelihood of major postoperative complications [127,143,144].

An additional challenge is the accurate confirmation of arterial wall involvement, which is often more difficult than venous assessment. The radiographic and intraoperative findings that are most specific for mural infiltration, such as luminal irregularity or focal caliber reduction, typically manifest late, whereas abutment or encasement alone does not reliably distinguish true wall penetration from close perivascular tumor or post-treatment fibrosis [21,22]. The arterial tunica media may further delay transmural extension, increasing the frequency of apparent arterial involvement without definitive mural disease. This diagnostic ambiguity raises concern for overtreatment, since some patients undergo formal AR with substantial morbidity despite final histopathology demonstrating no transmural arterial involvement [127]. These observations provide a rationale for artery-preserving approaches, most notably arterial divestment in carefully selected patients after NAT, when arterial adherence is suspected but true wall penetration remains uncertain [123,124]. Divestment seeks oncologically acceptable clearance through circumferential arterial skeletonization without reconstruction and may therefore reduce morbidity relative to formal AR [145,146]. Until higher-quality comparative data are available, transparent and standardized reporting of indications, operative technique, and perioperative and oncologic outcomes, together with rigorous patient selection, will be essential to better define benefit relative to risk and to inform evidence-based guideline development.

### Limitations

This review has several limitations. First, it was conceived as a narrative synthesis rather than a formal systematic review. Although a structured, predefined search strategy was applied, the absence of full systematic methods increases the possibility of incomplete literature capture and vulnerability to selection or publication bias. Second, the underlying evidence base is heterogeneous, with variability in definitions, NAT regimens, operative technique, perioperative pathways, and outcome reporting, which limits cross-study comparability and constrains causal inference. Third, some included reports enrolled mixed populations. While cohorts with a high proportion of pancreatectomies with concomitant vascular resections were excluded, residual confounding remains possible. Finally, the breadth of the topic required selective emphasis, and the lack of figures may reduce the clarity of proposed decision pathways.

## 8. Conclusions

Advancing the field will require rigorous patient selection and harmonized, a priori reporting of indications, operative technique, and prespecified oncologic and perioperative endpoints, thereby enabling reproducible comparisons and a more defensible estimation of benefit relative to risk. In curative-intent PDAC surgery, the overriding objective remains an R0 resection within a multidisciplinary treatment strategy. When vascular involvement jeopardizes margin negativity, vascular resection and reconstruction should be considered an enabling component of oncologic resection when technically feasible and oncologically justified, rather than dismissed a priori. Given that outcomes depend on operative expertise, coordinated perioperative care, and reliable longitudinal follow-up, these procedures are best centralized to specialized, high-volume hepatopancreatobiliary centers where consistent practice and transparent reporting can refine selection criteria, standardize techniques, and inform guideline development.

## Figures and Tables

**Figure 1 diagnostics-16-00102-f001:**
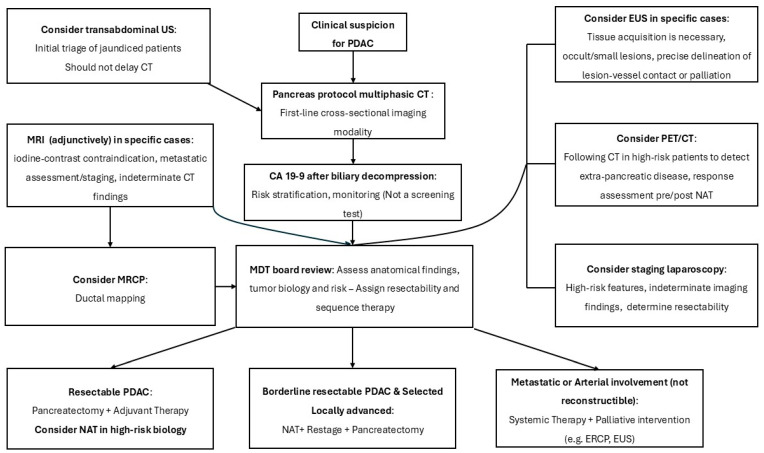
Diagnostic algorithm for PDAC with vascular involvement; PDAC = pancreatic ductal adenocarcinoma; NAT = neoadjuvant therapy; MDT = multidisciplinary tumor board; MRI = magnetic resonance imaging; CA 19-9 = carbohydrate antigen 19-9; EUS = endoscopic ultrasound; CT = computed tomography; PET/CT = positron emission tomography; ERCP = endoscopic retrograde cholangiopancreatography; high-risk biology = elevated CA 19-9, large tumor size, suspicious lymph nodes.

**Table 1 diagnostics-16-00102-t001:** Summary table of diagnostic modalities in PDAC assessment; CT = computed tomography; MRI = magnetic resonance imaging; MRCP = magnetic resonance cholangiopancreatography; FDG-PET = fluorodeoxyglucose positron emission tomography; PDAC = pancreatic ductal adenocarcinoma; NAT = neoadjuvant therapy; MPD = major pancreatic duct; PV = portal vein; SMV = superior mesenteric vein; SMA = superior mesenteric artery; CHA = common hepatic artery; CA = celiac axis/artery.

Modality	Use When	Findings	Limits	Role
Transabdominal ultrasound	Might be the first imaging modality in jaundiced patients (initial triage)	Biliary dilation and mass effect when acoustic window is favorable	Operator-dependent and limited by habitus and bowel gas	Generally limited
Pancreas-protocol CT	Suspected PDAC and initial diagnosis	Hypoenhancing mass, MPD cutoff with distal atrophy, double duct, vessel mapping PV SMV SMA CHA CA	Small or isoattenuating lesions may be missed, and post-NAT fibrosis can obscure planes while pancreatic fatty degeneration may further mask subtle tumor foci	Backbone for diagnosis and operative planning
Endoscopic ultrasound (EUS)	Histologic confirmation and evaluation of CT-occult lesions, with robust assessment of tumor–vascular involvement and potential utility for palliation (e.g., pain control)	Fine detail of small or isoattenuating lesions. Can show vascular invasion especially when combined with ancillary techniques	Invasive and operator-dependent	Adjunctive
MRI ± MRCP	Contrast contraindication or low lesion conspicuity on CT, with added value for metastatic assessment and ductal mapping	Ductal anatomy and subtle ductal cutoffs with higher sensitivity than CT for occult liver metastases	Longer acquisition time with susceptibility to motion artifacts and variable availability, with no demonstrated added value for assessing vascular invasion	Adjunctive
FDG-PET/CT	Before and after NAT in patients with high-risk biology or indeterminate extrapancreatic findings	Assessment of response to NAT with whole-body survey for extrapancreatic disease	Not a substitute for high-quality pancreas-protocol CT in PDAC, limited for detecting small-volume metastases, susceptible to false positives with inflammation and false negatives in low-FDG-avid disease, and associated with higher cost, variable availability, and additional radiation exposure	Adjunctive
Staging laparoscopy	High-risk features or indeterminate findings with assessment of resectability	Occult peritoneal or surface liver metastases	Invasive and operator-dependent	Adjunctive

**Table 2 diagnostics-16-00102-t002:** Venous operability and imaging criteria by NCCN 2025; NAT = neoadjuvant therapy.

Item	Definition or Threshold	Imaging Hallmarks on CT	Surgical Implication
Circumferential contact	Abutment ≤ 180° Encasement > 180°	Focal narrowing, Contour irregularity, Segmental occlusion, Intraluminal thrombus, Collateral veins	Further imaging findings must be considered to decide on treatment protocol
Resectable	No tumor contact or abutment without venous contour irregularity	Smooth vein outline, no focal narrowing, no thrombus	Standard pancreatectomy No or limited venous repair
Borderline resectable	Abutment or encasement with focal narrowing or contour irregularity or short-segment thrombosis when suitable proximal and distal targets exist	Teardrop sign or other luminal-narrowing deformity, Short-segment thrombus, Preserved targets for reconstruction	Consider NAT and reassess Plan venous repair Lateral venorrhaphy or patch or end to end if feasible
Locally advanced	Complete occlusion of SMV or PV not amenable to primary reconstruction at presentation	Long-segment occlusion, Extensive collaterals, Loss of suitable targets	Not operable at presentation Consider NAT and reassess

**Table 3 diagnostics-16-00102-t003:** Arterial criteria by location and imaging by NCCN 2025; CA = celiac axis/artery; SMA = superior mesenteric artery; CHA = common hepatic artery; HA = hepatic artery; CT = computed tomography.

Location	Resectable	Borderline Resectable	Locally Advanced	Imaging Predictors of Invasion
Head or uncinate	No measurable tumor contact with CA SMA or CHA	Short-segment abutment of CHA without extension to CA or to the HA bifurcation Abutment of SMA or limited abutment of a variant HA when safe reconstruction is feasible	Encasement of SMA or CA	Arterial encasement on high quality CT Specificity approaches 100% Sensitivity near 80% Ancillary signs include focal narrowing contour irregularity and segmental occlusion
Body or tail	No measurable contact with CA SMA or CHA	Abutment of CA when reconstruction or divestment is feasible at high volume centers	Encasement of SMA or CA Simultaneous CA and aortic involvement	Apply the same CT signs Use degree of contact in degrees Assess for loss of fat plane and arterial deformity

**Table 4 diagnostics-16-00102-t004:** Therapeutic rationale of PDAC; PDAC = pancreatic ductal adenocarcinoma; NAT = neoadjuvant therapy; MDT = multidisciplinary tumor board; CA 19-9 = carbohydrate antigen 19-9.

PDAC Category	Initial Management	Follow-Up Management	Operative Management	Notes
Resectable without high-risk biology	Upfront pancreatectomy at a high-volume center	Adjuvant chemotherapy after surgery	Proceed directly to resection	FOLFIRINOX is preferred for medically fit patients, with a gemcitabine-based regimen as an alternative, to complete a total of 6 months of systemic therapy, and management should be coordinated through an MDT
Resectable with high-risk biology	NAT	Assess treatment response using protocolized cross-sectional imaging and CA 19-9 monitoring	If no disease-progression is confirmed, consider diagnostic laparoscopy, then proceed to resection	Perioperative NAT is administered for 2 to 4 months preoperatively and 2 to 4 months postoperatively, with regimen selection tailored to patient fitness
Borderline resectable				Assess patients in an MDT setting and consider surgery with vascular resection at experienced centers when an R0 resection is achievable
Locally advanced	NAT and consider radiotherapy if no metastases are identified at diagnosis			
Indicators of high-risk biology	Use these features to guide suitability for NAT (elevated CA 19-9, suspicious nodes, large tumors)	—	—	Always assess within multidisciplinary setting

## Data Availability

Not applicable.

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
