# Peer review of "Diagnosis and Surgical Management for Advanced Pancreatic Cancer Requiring Vascular Resection"

_diagnostics, 2025, doi:10.3390/diagnostics16010102_

Round 1

Reviewer 1 Report

Comments and Suggestions for Authors

THe paper is well written and on a cutting-edge topic. These are my comments:

1) Please provide a reproducible search strategy with the string (not only the search terms) and with a flow chart of the studies included and excluded with reasons

2) The authors just briefly mentioned the role of elastography in EUS but i think the role of ancillary techniques should be commented deeper. Moreover, the authors should mention the role of contrast-enhanced EUS both in the staging/diagnosis and to guide tissue sampling (cite the relevant Srma: PMID: 33481633)

3) The authors should also mention the potential therapeutic role of EUS (mainly in palliation) in these patients (cite the relevant series PMID: 27356212)

4) I am not sure the paragraph on antithrombotic management is relevant here. For sure, it should be at least shortened.

5) Some surgical or radiological/EUS images would improve the quality of the manuscript

Author Response

We would like to express our sincere gratitude to the reviewer for the time and effort devoted to evaluating our work. We also thank the reviewer for their valuable, constructive comments and for sharing their expertise, which has helped us strengthen the quality, clarity, and scientific impact of the manuscript. Below, we provide our point-by-point response to the reviewer’s comments.

Comment 1

Please provide a reproducible search strategy with the string (not only the search terms) and with a flow chart of the studies included and excluded with reasons

Response

We thank the reviewer for this constructive suggestion. In response to this comment, we have expanded the description of our workflow in the Materials and Methods section and have provided the full, reproducible database-specific search strings, together with a graphical flow chart detailing study inclusion and exclusion with reasons, in the Supplementary Material. At the same time, we would like to clarify that the present work is a narrative review rather than a systematic review; Accordingly, it was not conducted under PRISMA guidelines. The literature search was therefore performed in a more flexible, topic-driven manner. Specifically, two authors independently searched two databases (PubMed and Scopus, respectively). Each author compiled relevant records, and final inclusion was determined through discussion and consensus among all authors.

Comment 2

The authors just briefly mentioned the role of elastography in EUS but I think the role of ancillary techniques should be commented on deeper. Moreover, the authors should mention the role of contrast-enhanced EUS both in the staging/diagnosis and to guide tissue sampling (cite the relevant Srma: PMID: 33481633).

Response

We would like to thank the reviewer for this valuable suggestion and for highlighting this very interesting and well conducted meta-analysis (PMID 33481633). We acknowledge that the discussion of ancillary EUS techniques was previously too brief and that a deeper commentary is warranted. We agree that the role of contrast-enhanced EUS should be explicitly addressed both in diagnosis/staging and in guiding tissue sampling. We believe that the addition of the suggested meta-analysis significantly enhances the quality of the manuscript. These points have now been implemented in the revised version, and the relevant study has been cited accordingly (lines 128-131).

Comment 3

The authors should also mention the potential therapeutic role of EUS (mainly in palliation) in these patients (cite the relevant series PMID: 27356212)

Response

We would like to thank the reviewer for their constructive suggestion and for pointing to the relevant study (PMID: 27356212). We agree that the initial version did not sufficiently address the potential therapeutic role of EUS, particularly its palliative applications in this complex patient population. We consider that incorporating this evidence adds important clinical context and further improves the completeness and practical relevance of the manuscript. This aspect has now been integrated into the revised text, with appropriate citations of the suggested study (lines: 135-141).

Comment 4

I am not sure if the paragraph on antithrombotic management is relevant here. For sure, it should be at least shortened.

Response

We would like to thank the reviewer for this constructive and valuable suggestion. We acknowledge that the antithrombotic management is not a central element of the present manuscript and does not constitute a primary objective of the review. Accordingly, the relevant section has been substantially shortened, with the focus retained only on the most clinically pertinent points (551-588).

Comment 5

Some surgical or radiological/EUS images would improve the quality of the manuscript

Response

We express our sincere gratitude to the reviewer for this comment and valuable suggestion. We carefully considered incorporating representative radiological, EUS, and intraoperative figures to further enhance the manuscript. However, given the breadth of the topic and the number of distinct technical and clinical domains covered, a limited selection of images would risk disproportionately emphasizing certain sections while underrepresenting others. In addition, providing an appropriately comprehensive and balanced set of figures would require a substantial number of images, which is not feasible within the current revision timeline. We acknowledge this as a limitation of the present study, and we have therefore stated this explicitly in the Limitations section (lines 699-710).

Reviewer 2 Report

Comments and Suggestions for Authors

This is a well-written and very comprehensive narrative review on the diagnosis and surgical management of PDAC with vascular involvement. The authors clearly put a lot of work into summarizing a very broad area. Still, there are places where the manuscript becomes overly long, repetitive, and a bit unfocused for a narrative review. Some sections feel more like a textbook chapter than a diagnostic/surgical management review. A bit of tightening and clarifying would help the flow and make the key points easier to follow. It’s a timely topic, and the paper is generally clear and comprehensive. I thank the authors for putting this together.

MAJOR COMMENTS

1. The manuscript is extremely long and reads more like a textbook chapter than a focused review. The level of detail (especially in diagnostic modalities, CT/MRI/EUS performance data, and venous reconstruction techniques) is beyond what is usually expected in a narrative review for a clinical diagnostics journal. Consider shortening significantly—many paragraphs repeat similar concepts, especially CT vs MRI accuracy and NAT pathways. The message gets diluted.

2. The stated aim is a narrative review, but the methods section almost mimics a systematic review without the structure. You list search terms but not inclusion strategy, timeframe, or justification for data selection. For a narrative review this doesn’t need to be elaborate, but right now it feels half-systematic and half-narrative. Clarify that this is not a systematic review, or expand the methodology slightly so it is consistent with what you present. Even without PRISMA, readers need a clearer sense of how the literature was selected and what was excluded.

3. Several tables (especially Tables 1–7) are overly dense, duplicative, or contain textbook-level detail not necessary for the target audience.
For example, the diagnostic tables repeat in slightly different forms; venous patency details across multiple studies could be condensed to one paragraph; and Table 7 does not really add much beyond what the narrative already says. Consider trimming or merging to reduce redundancy.

4. The review lacks critical discussion in some areas and reads mostly descriptive.
For instance, arterial resection: the manuscript describes techniques and meta-analysis outcomes in depth, but there is limited critical comment on why outcomes remain poor, patient selection challenges, and ongoing controversies. Adding more commentary and your own interpretation would strengthen the review.

5. Neoadjuvant therapy section is very long and could be streamlined.
You cite many excellent trials, but the result is fragmented and sometimes repetitive. A synthesized paragraph summarizing:

when NAT is preferred,

typical duration,

evidence hierarchy (FOLFIRINOX vs gemcitabine-based),

practical markers for response
would make the section much easier to read.

6. The review occasionally shifts into procedural detail unsuitable for a narrative review format.
Long paragraphs describing venous graft choices, patch materials, step-by-step anastomotic preferences, or SMA-first approaches are more fitting for a surgical techniques paper. These sections could be shortened to maintain focus.

Minor Comments:

1. The introduction is slightly heavy; could be shortened.

2. A couple of abbreviations first appear without being defined. Some typographical/grammatical slips. A few small issues (e.g., “mesenteriv,” “biologically high risk,” missing hyphens, repeated words). A careful language polish is needed.

Overall, this is a strong, well written and useful narrative review. With clearer methods description and some tightening, this will read much more smoothly.

Author Response

We would like to sincerely thank the reviewer for their thoughtful and thorough assessment of our manuscript, as well as for the time and expertise devoted to this evaluation. We are grateful for their constructive comments regarding length, repetition, and overall focus, which we found highly valuable. We have carefully considered each point and revised the manuscript to tighten the narrative, improve clarity, and enhance the flow while preserving the breadth of coverage appropriate for this topic. We appreciate the reviewer’s guidance, and we believe their suggestions have meaningfully strengthened the manuscript and elevated its scientific quality.

Comment 1

The manuscript is extremely long and reads more like a textbook chapter than a focused review. The level of detail (especially in diagnostic modalities, CT/MRI/EUS performance data, and venous reconstruction techniques) is beyond what is usually expected in a narrative review for a clinical diagnostics journal. Consider shortening significantly—many paragraphs repeat similar concepts, especially CT vs MRI accuracy and NAT pathways. The message gets diluted.

Response

We would like to thank the reviewer for this valuable and constructive comment. We agree that the initial version was overly extensive and that, in certain sections, the level of detail and repetition could compromise readability and obscure the principal messages. In accordance with the reviewer’s recommendation, we have substantially shortened the manuscript and reduced redundancy throughout. Specifically, we streamlined the sections on diagnostic modalities, including CT, MRI, and EUS performance data (lines 104–153). Subsection 3.1, “Diagnostic Algorithm–Summary,” has also been removed to avoid repetition. In addition, the “Therapeutic Protocol” subsection has been merged into Section 4, “Management of PDAC with Vascular Involvement,” and overlapping content related to neoadjuvant therapy pathways has been consolidated to improve coherence and narrative continuity (lines 230–320). We further condensed the technical description of venous and arterial reconstruction approaches (lines 419–460, 461–473, and 522–550) and shortened the discussion of antithrombotic practice to retain only the most clinically relevant considerations (lines 551–588).

Comment 2

The stated aim is a narrative review, but the methods section almost mimics a systematic review without the structure. You list search terms but not inclusion strategy, timeframe, or justification for data selection. For a narrative review this doesn’t need to be elaborate, but right now it feels half-systematic and half-narrative. Clarify that this is not a systematic review or expand the methodology slightly so it is consistent with what you present. Even without PRISMA, readers need a clearer sense of how the literature was selected and what was excluded.

Response

We would like to thank the reviewer for this insightful and constructive comment. Although we followed a predefined and structured approach to identify and organize the relevant literature, the search was not performed systematically and was not conducted in full adherence to PRISMA methodology. In response to the reviewer’s concern, we have revised and clarified the “Materials and Methods” section to explicitly state the narrative nature of the review and to better describe our literature identification and selection approach, including how sources were prioritized and which categories of publications were not considered (lines 62-78). In addition, to further improve transparency and provide readers with a clearer overview of our workflow, we have added a supplementary figure and accompanying supplementary material outlining the literature selection and synthesis process.

Comment 3

Several tables (especially Tables 1–7) are overly dense, duplicative, or contain textbook-level detail not necessary for the target audience. For example, the diagnostic tables repeat in slightly different forms; venous patency details across multiple studies could be condensed to one paragraph; and Table 7 does not really add much beyond what the narrative already says. Consider trimming or merging to reduce redundancy.

Response

We thank the reviewer for this careful and practical suggestion. In line with the reviewer’s recommendation, we have revised the tabular content to reduce redundancy and improve readability. Specifically, Tables 6 and 7 have been removed. Table 5 has been transferred to the supplementary material, as it provides supportive rather than essential information for the main narrative. The remaining tables have been compacted, harmonized to minimize overlap, and reviewed to ensure that they present only the most clinically relevant and high-yield data, thereby enhancing clarity and overall usefulness.

Comment 4

The review lacks critical discussion in some areas and reads mostly descriptive. For instance, arterial resection: the manuscript describes techniques and meta-analysis outcomes in depth, but there is limited critical comment on why outcomes remain poor, patient selection challenges, and ongoing controversies. Adding more commentary and your own interpretation would strengthen the review.

Response

We thank the reviewer for this important comment. We fully agree that, in the initial submission, several sections were more descriptive than interpretative, with insufficient critical appraisal of the underlying controversies and limitations in the current evidence base. We acknowledge that incorporating a more analytical perspective, particularly in challenging and debated domains such as arterial resection, would strengthen the manuscript and improve its value to the target audience. In response, we carefully considered the reviewer’s recommendation and revised the manuscript accordingly. Sections 3 through 7 have been enhanced with additional critical discussion and author interpretation, including commentary on why outcomes may remain suboptimal despite technical advances, the challenges of patient selection and risk stratification, and the major areas of ongoing debate. These additions aim to provide a more balanced synthesis that is not only technically informative but also clinically and conceptually useful for readers.

Comment 5

Neoadjuvant therapy section is very long and could be streamlined.
You cite many excellent trials, but the result is fragmented and sometimes repetitive. A synthesized paragraph summarizing: when NAT is preferred, typical duration, evidence hierarchy (FOLFIRINOX vs gemcitabine-based), practical markers for response would make the section much easier to read.

Response

We express our sincere gratitude to the reviewer for this thoughtful and highly constructive suggestion. We fully agree that the original neoadjuvant therapy section, although comprehensive, was overly extensive and, in parts, repetitive, which could detract from readability. In accordance with the reviewer’s recommendation, we revised this section substantially (lines 229-320). We removed overlapping content, improved narrative continuity, and added a concise synthesized paragraph summarizing the principal clinical takeaways, including when neoadjuvant therapy is preferred, typical treatment duration, the relative evidence supporting FOLFIRINOX- versus gemcitabine-based regimens, and practical markers used to assess treatment response. We believe these amendments enhance clarity while maintaining the rigor and completeness of the supporting evidence.

Comment 6

The review occasionally shifts into procedural detail unsuitable for a narrative review format. Long paragraphs describing venous graft choices, patch materials, step-by-step anastomotic preferences, or SMA-first approaches are more fitting for a surgical techniques paper. These sections could be shortened to maintain focus.

Response

We would like to thank the reviewer for this pertinent and constructive comment. We fully agree that certain portions of the manuscript, particularly within Section 5, contained an excessive level of procedural and step-by-step technical detail that is more appropriate for a dedicated surgical techniques paper rather than a narrative review. In accordance with the reviewer’s recommendation, we have shortened and revised the relevant passages, including those addressing venous graft selection, patch materials, anastomotic preferences, and SMA-first approaches (lines.369-375 & 461-473). These sections have been refocused to emphasize the overarching concepts, comparative considerations, and clinically relevant decision-making points, while avoiding unnecessary technical detail.

Comment 7

The introduction is slightly heavy; could be shortened.

Response

We are grateful to the reviewer for this valuable observation. We agree that the Introduction in the initial version was overly detailed and could appear heavy, which may reduce readability and delay the transition to the central aims of the review. Accordingly, we have abbreviated and refined the Introduction by removing nonessential background content and consolidating overlapping statements, while preserving the key context, rationale, and objectives of the manuscript (lines 39-61). We believe this revision improves clarity and provides a more focused entry into the main body of the review.

Comment 8

A couple of abbreviations first appear without being defined. Some typographical/grammatical slips. A few small issues (e.g., “mesenteriv,” “biologically high risk,” missing hyphens, repeated words). A careful language polish is needed.

Response

We appreciate the reviewer for noting these points. We agree that consistency in abbreviation use and careful language accuracy are essential for clarity and professionalism. In response, we performed a thorough revision across the manuscript. All abbreviations are now defined at first mention and used consistently thereafter. We also corrected typographical and grammatical errors, addressed repeated words, standardized hyphenation and terminology, and amended specific issues highlighted by the reviewer.

Round 2

Reviewer 1 Report

Comments and Suggestions for Authors

The revised manuscript is OK. Thank you!